# Clinical Features and Outcomes of VAP Due to Multidrug-Resistant *Klebsiella* spp.: A Retrospective Study Comparing Monobacterial and Polybacterial Episodes

**DOI:** 10.3390/antibiotics12061056

**Published:** 2023-06-15

**Authors:** Dalia Adukauskiene, Ausra Ciginskiene, Agne Adukauskaite, Despoina Koulenti, Jordi Rello

**Affiliations:** 1Medical Academy, Lithuanian University of Health Sciences, 44307 Kaunas, Lithuania; daliaadu@gmail.com; 2Department of Cardiology and Angiology, University Hospital of Innsbruck, 6020 Innsbruck, Austria; agne.adu@gmail.com; 3Second Critical Care Department, Attikon University Hospital, 12462 Athens, Greece; deskogr@yahoo.gr; 4UQ Centre for Clinical Research (UQCCR), Faculty of Medicine, The Univesrity of Queensland, 4029 Brisbane, Australia; 5Vall d‘Hebron Institute of Research, Vall d‘Hebron Campus Hospital, 08035 Barcelona, Spain; 6Clinical Research, CHU Nîmes, 30900 Nîmes, France

**Keywords:** ventilator-associated pneumonia, *Klebsiella*, multidrug-resistance, antibiotic treatment optimisation, aspiration pneumonia, hospital-acquired pneumonia, antibiotic stewardship, non-fermentative GNB, pneumonia resolution, mortality, sepsis, polymicrobial

## Abstract

VAP due to multidrug-resistant (MDR) bacteria is a frequent infection among patients in ICUs. Patient characteristics and mortality in mono- and polybacterial cases of VAP may differ. A single-centre, retrospective 3-year study was conducted in the four ICUs of a Lithuanian referral university hospital, aiming to compare both the clinical features and the 60-day ICU all-cause mortality of monobacterial and polybacterial MDR *Klebsiella* spp. VAP episodes. Of the 86 MDR *Klebsiella* spp. VAP episodes analyzed, 50 (58.1%) were polybacterial. The 60-day mortality was higher (*p* < 0.05) in polybacterial episodes: overall (50.0 vs. 27.8%), in the sub-group with less-severe disease (SOFA < 8) at VAP onset (45.5 vs. 15.0%), even with appropriate treatment (41.7 vs. 12.5%), and the sub-group of extended drug-resistant (XDR) *Klebsiella* spp. (46.4 vs. 17.6%). The ICU mortality (44.0 vs. 22.5%) was also higher in the polybacterial episodes. The monobacterial MDR *Klebsiella* spp. VAP was associated (*p* < 0.05) with prior hospitalization (61.1 vs. 40.0%), diabetes mellitus (30.6 vs. 5.8%), obesity (30.6 vs. 4.7%), prior antibiotic therapy (77.8 vs. 52.0%), prior treatment with cephalosporins (66.7 vs. 36.0%), and SOFA cardiovascular ≥ 3 (44.4 vs. 10.0%) at VAP onset. Patients with polybacterial VAP were more likely (*p* < 0.05) to be comatose (22.2 vs. 52.0%) and had a higher SAPS II score (median [IQR] 45.0 [35.25–51.1] vs. 50.0 [40.5–60.75]) at VAP onset. Polybacterial MDR *Klebsiella* spp. VAP had distinct demographic and clinical characteristics compared to monobacterial, and was associated with poorer outcomes.

## 1. Introduction

Patients treated in intensive care units (ICU) have a higher risk of infections, due to increased requirements for invasive devices, expressed selective antibacterial pressure, and significant immunosuppression because of the critical illness. Despite improved preventive efforts, ventilator-associated pneumonia (VAP) remains the most common ICU infection of patients on mechanical ventilation (MV), and is associated with substantial morbidity/mortality, a longer length of stay (LOS) in the hospital, and also increased healthcare costs, especially in VAP caused by multidrug-resistant (MDR) bacteria [1,2,3,4].

Based on the data from recent multicentre European studies, the most common (80%) VAP pathogens were *Pseudomonas aeruginosa (P. aeruginosa)*, *Acinetobacter baumannii (A. baumannii)*, and *Klebsiella pneumoniae* (*K. pneumoniae*) [5,6]. Moreover, up to 70% of all hospital-acquired infections (HAI) were caused by Gram-negative bacteria (GNB), with *Klebsiella* spp. strains representing up to 10% of them [5]. This threatening pathogen of VAP, accounting for around 10% of hospital-acquired pneumonia (HAP) and VAP cases, is associated with a constantly rising rate of multidrug resistance [7]. In 2022, data derived from the Lithuanian Institute of Hygiene revealed that *Klebsiella* spp. was the dominating (20%) pathogen in pneumonias of ICU patients [8].

The increasing incidence of *Klebsiella* spp. infections and the emergence of antibiotic-resistant strains have brought this pathogen to the forefront of attention [9,10]. According to the 2019 data of EARS-NET, more than a third (36.6%) of *Klebsiella* spp. strains reported in the EU/EEA were found to be resistant to one or more of the antibacterial groups under surveillance (third-generation cephalosporins, fluoroquinolones, aminoglycosides, and carbapenems) [11]. The rate of MDR isolates of *Klebsiella* spp. varies between countries, predominating in Eastern and South-Western Europe. In Mediterranean countries, the resistance of *Klebsiella* spp. strains to the third-generation cephalosporins, fluoroquinolones, and aminoglycosides has exceeded 65%. The resistance of *Klebsiella* spp. to carbapenems has increased seven-fold since 2006 [12]. Consequently, extended-spectrum beta-lactamases (ESBL) producing *Enterobacterales*, including *Klebsiella* spp., as well as carbapenem-resistant *A. baumannii* and *P. aeruginosa*, have been recognized by the WHO as major challenges for the future [13].

Previous studies indicate that the outcomes of VAP patients depend not only on patient-related factors (demographic data, comorbidities, socioeconomic conditions) and disease severity, but also on the specific causative bacterium, its antibiotic resistance, and virulence [14,15]. Yet, the point of view that MDR pathogens are associated with higher mortality, remains controversial until now. Numerous studies have confirmed this association in settings of various infections [1,16,17,18]. The published data show that nearly a quarter of VAP cases are of polybacterial origin [19,20]. Critical illness-induced immunosuppression is thought to increase the risk of polybacterial pulmonary infections. Early administration of appropriate antibiotic therapy in the cases of polybacterial infections may be challenging, because the co-pathogens may have different susceptibilities to antibiotics. There is a scarcity of scientific studies that examine the distinctions between polybacterial and monobacterial infections. Therefore, it remains unclear whether the polybacterial origin of infection is associated with increased disease severity and worse patient outcomes [1,16,17,18,21]. More than two decades ago, a 1.5-year-duration study in French ICUs found no differences in mortality of mono- and polybacterial VAP [19]. However, a recent study by Karakonstantis et al. on pulmonary and bloodstream infections (BSI) has shown that the type of pathogen isolation (monobacterial or polybacterial) may be associated with outcomes [22]. Infections due to MDR *Klebsiella* spp. were found to be associated with high mortality and hospital outbreaks [18,23]. Despite the clinical relevance of *Klebsiella* spp., there is a considerable gap in knowledge of differences in outcomes and clinical features of mono- vs. polybacterial cases of VAP, due to MDR *Klebsiella* spp. No comparable multicentre studies have been published.

The hypothesis of our study is that mortality and clinical characteristics of patients with VAP due to MDR *Klebsiella* spp. differ between monobacterial and polybacterial cases. A potential approach to evaluating these differences is to compare the outcome of patients with monobacterial VAP to those with polybacterial. Time points that are too early might not capture late mortality, due to infection-related complications. For that reason, the primary objective of the current study was to compare the all-cause 60-day mortality between mono- and polybacterial episodes of VAP due to MDR *Klebsiella* spp. The secondary objective was to compare the ICU mortality and various clinical aspects of mono- and polybacterial VAP due to MDR *Klebsiella* spp.

## 2. Materials and Methods

A secondary analysis of a retrospective cohort study of medical records of patients that were admitted to the 72-bed adult ICU (medical-surgical, neurosurgical, cardiosurgical, and coronary) was conducted at Lithuania’s largest (2300-bed) university-affiliated hospital over a three-year period (from January 2014 to December 2016). The study was approved by the Kaunas Regional Biomedical Research Ethics Committee (No. BE-2-13). The need for written consent was waived, due to the study’s observational nature.

Inclusion criteria were as follows: (1) age ≥ 18 years, (2) the first episode of VAP due to MDR *Klebsiella* spp. Pneumonia was defined as ventilator-associated when it occurred after 48 or more hours after endotracheal intubation and the onset of mechanical ventilation. The clinical diagnosis of VAP was made according to the 2005 ATS/IDSA criteria [24]. All tracheal aspirate samples were assessed for purulence, and only bacteria demonstrating significant and moderate growth in the tracheal aspirate culture (using a conventional semi-quantitative method) were considered as causative pathogens of VAP. *Candida* spp. and *Enterococci* were not regarded as VAP pathogens. The identification of *Klebsiella* isolates and antibiotic susceptibility was performed according to the EUCAST guidelines [25]. *Klebsiella* isolates were defined as MDR according to an international expert proposal for the interim standard definitions for acquired resistance criteria [26]. The detailed case enrolment is presented in Figure 1.

Demographics, clinical and laboratory data collected for each VAP case included: (1) the data of collection of the first MDR *Klebsiella* spp. positive tracheal aspirate (TA) culture and drug resistance of *Klebsiella* spp. strains; (2) age, gender, type of admission (medical/surgical), comorbidities, and the use of intravenous (IV) antibiotics within the previous 90 days; (3) concomitant infections, (4) red blood cell (RBC) transfusion, reintubation, tracheostomy, and the need for renal replacement therapy (RRT) during the ICU stay; (5) sepsis status, oxygenation, temperature, inflammatory and acid-base balance on VAP diagnosis; (6) severity of illness on ICU admission and VAP onset; and (7) outcome: alive or dead at day 60, and also ICU discharge. The SOFA score was used to define organ dysfunction (>0) and organ failure (>2). Sepsis status was diagnosed according to Sepsis-2 criteria [27]. The severity of illness was assessed on ICU admission and the diagnosis of VAP using the Sequential Organ Failure Assessment (SOFA) and Simplified Acute Physiology Score (SAPS) II scores; data were reported as the worst values within 24 h. Comorbidities were assessed using the Charlson Comorbidity Index (CCI). Admission was considered as surgical in patients who had undergone surgery in the preceding four weeks. A body mass index of more than 30 was defined as obesity. Non-survivors during the first 24 h of ICU admission were excluded from the mortality analysis. The 60-day and ICU mortality were defined as all-cause mortality within 60 days or during the ICU stay after VAP diagnosis, respectively. Those who remained alive at day 60 or during the ICU stay were considered to be survivors.

To estimate the effect of confounders that may influence mortality, mono- and polybacterial VAP cases we stratified into subgroups according to the severity of illness (SOFA score > 7 vs. SOFA score < 8), the appropriateness (appropriate vs. inappropriate) and timeliness (early vs. late) of the antibiotic therapy, and the MDR profile (MDR vs. XDR). The treatment was considered appropriate when the patient received at least one antibiotic active in vitro against all identified bacteria suspected as pathogens. Early definitive antibacterial treatment was defined as administering at least one appropriate antibiotic within 48 h from the pathogen identification.

### Statistical Analysis

Categorical variables were summarized as frequencies and percentages [*n* (%)] and continuous variables as medians and interquartile range [IQR]. The Mann–Whitney non-parametric test, Pearson’s chi-square test, or two-tailed Fisher’s exact test were performed to detect the differences between groups, as appropriate. Mortality was analyzed as a binary outcome (survivor/non-survivor) and survival time data. In the survival analysis, Kaplan–Meier estimates of the probability of survival were obtained, and survival curves were compared between groups using Log Rank. In all analyses, two-sided *p* values of <0.05 were considered to be statistically significant. The G*Power 3.1 program was used to estimate a power of study. Statistical analysis was carried out while using the Statistical Package for the Social Sciences SPSS version 20 (SPSS, Chicago, IL, USA).

## 3. Results

A total of 86 VAP cases due to MDR *Klebsiella* spp. (*K. pneumoniae, n* = 82, *K. oxytoca*, *n* = 4) were identified: 36 (41.9%) monobacterial and 50 (58.1%) polybacterial. All MDR *Klebsiella* spp. strains of the cohort were susceptible to carbapenems, but the vast majority of them (>95%) were resistant to piperacillin, ampicillin-sulbactam, and cephalosporins. More detailed data on the antibacterial resistance of *Klebsiella* spp. strains are shown in Figure 2. Combined resistance to penicillin, third-generation cephalosporins, aminoglycosides, and fluoroquinolones was found in 42 strains (48.8%).

In association with MDR *Klebsiella* spp., one co-pathogen was found in forty-one (80.0%), and two co-pathogens in nine (20.0%) cases of polybacterial VAP. The most frequently isolated co-pathogens were *P. aeruginosa* (*n* = 20, 20% carbapenem resistant) and *A. baumannii* (*n* = 12, 91.7% carbapenem resistant). All other Gram-negative co-pathogens (*E. coli*, *n* = 7, *Citrobacter* spp., *n* = 4, *Proteus* spp., *n* = 2, *Enterobacter* spp., *n* = 1) were susceptible to carbapenems. Among the *Staphylococcus aureus* co-pathogens, only one (11.1%) was MRSA.

*Klebsiella* spp. VAP episodes were, in the vast majority, late-onset, occurring at a median 13.0 days (IQR 6.0–20.0) after hospitalization and 8.0 days (IQR 5.0–13.3) after intubation. The episodes occurred more frequently in males (88.4%), in elderly patients (median 67.0 years, IQR 55.0–75.3), and patients admitted to the ICU with high severity of illness (median SAPS II and SOFA of 47.0 and 6.0, and IQR 35.0–55.0 and 4.8–9.0, respectively). Fifty-six (65.1%) were postoperative adults. Five (5.8%) patients had COPD, thirteen (15.1%) cancer, sixteen (18.6%) diabetes and fifteen (17.4%) obesity (body mass index above 30). Seventeen (19.8%) of MDR *Klebsiella* spp. VAP cases were in the medical-surgical ICU, 37 (43.0%) in the neurosurgical ICU, six (7.0%) in the coronary care unit, and twenty-six (30.2%) in the cardiosurgical ICU. At diagnosis, twenty (23.3%) of the patients underwent a tracheostomy. In twelve (14%) cases, *Klebsiella* spp. was isolated both from blood and tracheal aspirate cultures. No empyema or necrotizing pneumonia was diagnosed. Twenty-eight (32.6%) out of eighty-six patients had a previous (within 7 days before the onset of VAP) or concomitant infection. Twenty-one (24.4%) patients required vasopressors at VAP diagnosis, and eighteen (20.9%) underwent renal replacement therapy during their ICU stay. The median PaO2/FiO2 at diagnosis was 162.5 (IQR 126.5–224.2) mmHg. Median CRP and WBC were 155.5 mg/L (IQR 112.3–231.5) and 13.1 × 10^9^/L (IQR 9.9–17.6), respectively. A comparison of clinically significant differences between mono- and polymicrobial episodes is detailed in Table 1.

Patients with monobacterial episodes more frequently had diabetes, obesity, prior antibiotic use (particularly cephalosporins), prior hospitalization within 90 days before VAP onset, a cardiovascular SOFA score ≥ 3, and septic shock at VAP onset. Polybacterial episodes were found to be associated with higher SAPS II scores and altered consciousness (as depicted by neurological SOFA score ≥ 3) at VAP onset (detailed comparison in Table 1). No differences were found between monobacterial and polybacterial episodes in the rates of disease severity (SOFA and SAPS II scores) and organ failure on ICU admission, of CCI > 3, of reintubation, or of RBC transfusion prior to VAP. The possible variables of infection severity were comparable between mono- and polybacterial cases, including the rates of sepsis, organ failure, hypoxia severity, hyper- or hypothermia, acidosis, and elevation of inflammatory markers (CRP, leukocytosis) at the onset of VAP.

Overall, the all-cause 60-day mortality was 40.7% (35/86), while the 60-day all-cause mortality of polymicrobial cases was 50% (25/50). Patients with polybacterial VAP due to MDR *Klebsiella* spp. had an estimated excess all-cause 60-day mortality of 23.2% compared to those with monobacterial VAP. ICU mortality was 44.0% vs. 22.8% (*p* = 0.028), respectively. The time to death (censored at day 60) was found to be shorter in the group of polybacterial VAP, due to MDR *Klebsiella* spp., *p* = 0.046 (Figure 3).

The all-cause mortality rate within 60 days for cases of polybacterial VAP tended to be higher, irrespective of previous or concomitant infection. Additionally, in cases of monobacterial VAP with concomitant infection, the mortality rate was 14.3% vs. 52.4% in polybacterial cases (*p* = 0.078). Mortality remained higher in polybacterial cases, regardless of the timeliness of appropriate definitive treatment (early vs. late), although the difference was not statistically significant. The detailed characteristics of subgroups in all-cause 60-day mortality are provided in Table 2.

## 4. Discussion

The current study on VAP due to MDR *Klebsiella* spp. added value to the understanding of its epidemiological features and the impact of the monobacterial vs. polybacterial origin of VAP on patient mortality. Two-thirds of the episodes were polymicrobial with an all-cause 60-day mortality of 50%. An important new finding of our analysis was that the 60-day mortality rate was 20% higher in polybacterial compared to monobacterial cases. The trend of higher mortality of polybacterial cases persisted even after adjusting for the impact of disease severity, appropriateness and timeliness of treatment, and the resistance profile of *Klebsiella* spp. strains. Additionally, notable differences were found in the demographic and clinical characteristics of mono- and polybacterial cases of VAP due to MDR *Klebsiella* spp.

During recent decades, progress in diagnostic techniques has aided in detecting the rising prevalence of polybacterial pneumonia [28,29]. Notwithstanding this, the studies examining the discrepancies in outcomes among patients with mono- vs. polybacterial VAP caused by the same pathogen are very limited [30,31]. Despite the increasing knowledge about the composition of the normal microbiome of the lower respiratory tract, previously thought to be sterile, it is still difficult to clarify the precise role of each microorganism in polybacterial lung infections: are they true etiological pathogens or merely colonizers/commensals?

Previous research showed high (up to 57.1%) all-cause mortality when VAP was caused by hypervirulent *K. pneumoniae*; however, it remains unclear whether, and to what degree, the mortality can be associated with the pathogenicity and antibiotic resistance of *Klebsiella* spp. strains, the underlying comorbidities, disease severity (organ dysfunction), or presence of co-pathogens in the case of polybacterial infections [7,32]. The impact of a specific pathogen on the disease course and outcome in the case of polybacterial infections may be difficult to estimate because of the possibility of either positive or negative bacterial interactions. Although the prevalence of *Klebsiella* spp. as a VAP-causing pathogen is on the rise in several countries, so far we have found no study comparing the impact of a monobacterial and polybacterial origin for VAP due to MDR *Klebsiella* spp. on patient mortality. The current study expands the body of knowledge, demonstrating that the polybacterial VAP cases had significantly higher all-cause 60-day mortality (50%) compared to the monobacterial *Klebsiella* spp. VAP (27.8%). Interestingly, our earlier study (mono- vs. polybacterial VAP due to MDR *A. baumannii*) found significantly higher mortality for monobacterial cases (57.1% vs. 37.3%) [31]. In the study by Brewer et al. (mono- vs. polybacterial VAP due to *P. aeruginosa*), a similar trend (78.0% vs. 53.0%, *p* = 0.15) was observed [30]. On the other hand, Combes et al. did not identify any significant differences in mortality rates of patients with mono- vs. polybacterial VAP [19]. Nevertheless, it should be highlighted that it is difficult to make a comparison between our findings and those of Combes et al., as they did not investigate the mono- vs. polybacterial VAP cases caused by the same pathogen. Furthermore, the *Klebsiella* species accounted only for 3.1% of all pathogens, and caused monobacterial cases only. Yet, there are studies of pneumonia that, similar to our findings, have revealed higher mortality in polybacterial cases [21,28,33].

After adjusting for the severity of illness, we observed a statistically significantly increased all-cause 60-day mortality for polybacterial VAP cases in the less-severe (SOFA < 8) disease group. In a recent study by Chang et al. on HAP/VAP in ICU patients (*Klebsiella* spp. was the third most-common pathogen), an association between reduced 28- and 60-day mortality and both the appropriate empirical and definitive treatment was found [34]. Nonetheless, it is noteworthy that, contrary to our study, in the study by Chang, only 67% of pathogens were MDR bacteria, and only 12% of all HAP/VAP cases were of polybacterial origin. Additionally, the associations of the pneumonia‘s origin, appropriateness of treatment, and mortality were not tested. The appropriateness and timeliness of administered antibacterial therapy might have significant implications on patient mortality. Data from the large multicentre study by Kadri et al. (2021) have shown an association between inappropriate empirical antibiotic therapy in monobacterial bloodstream infections and increased mortality [35]. We analysed the possible impact of both the appropriateness and timeliness of definitive treatment on the mortality of mono- vs. polybacterial cases of VAP due to MDR *Klebsiella* spp. The time of appropriate therapy (early vs. late) was not significantly associated with mortality differences in mono- vs. polybacterial VAP cases. In the early-and-appropriate-treatment sub-group, an average 20% excess 60-day mortality (40.0% vs. 21.7%) in polybacterial VAP cases was still documented. This excess mortality could be partly explained by the fact that, when pneumonia is caused by multiple MDR bacteria, it might be challenging or even impossible to achieve a minimal inhibitory concentration of antibiotics effective against all pathogens. Furthermore, not all parenterally administered antibiotics effectively diffuse into the lung tissue, so their impact on the same pathogen in the bloodstream and lung tissue may vary. The analysis of the combined effects of disease severity and appropriateness of treatment revealed a statistically significant increase in the 60-day mortality rate in polybacterial VAP cases in the sub-group with the lower disease severity (*p* = 0.049). Our finding that the polybacterial MDR *Klebsiella* spp. VAP had a poorer outcome compared to the monobacterial cases might, at least partially, be explained by the possible synergism between multiple pathogens leading to the activation of virulence mechanisms, the alterations of infected niche, or an imbalanced host–pathogen response [29,36,37]. The cooperative influence of multiple bacteria on the disease course can be more deleterious than the effect of each bacterium independently. The intricate interplay between bacteria encompasses metabolite exchange, interspecies quorum signals, co-aggregation, biofilm formation, and the production of intrinsic antimicrobials [38]. These interactions are key factors in promoting the pathogenesis of most co-pathogens. Moreover, the study by Liao et al. has shown that beta-lactamase-producing bacteria can indirectly contribute to pathogenesis by shielding other bacteria in polybacterial infection cases [39]. The susceptibility of co-pathogens to antibiotics may also have an important impact on the outcomes of patients with polybacterial VAP. As shown by the study of polybacterial bacteremia by Wervay et al., attributable mortality may vary, depending on the causative pathogen [40]. These findings indicate that the influence of polybacterial infections on outcomes should be investigated separately according to the causative pathogens, as was carried out in our recent study on MDR *A. baumannii* and the current study on MDR *Klebsiella* spp. VAP. The resistance of *Klebsiella* spp. strains to antibiotics used to treat infections caused by these bacteria is constantly rising [11,41]. Corresponding to previous studies, we found MDR *Klebsiella* spp. strains to be fully or nearly fully (>97%) resistant to piperacillin, second- and third-generation cephalosporins, and ampicillin/sulbactam [12,42,43,44]. Additionally, we observed a very high percentage (>50%) of resistance to gentamycin, ciprofloxacin, and piperacillin/tazobactam. Similarly, as in the study from neighbouring Poland, we found all MDR *Klebsiella* spp. strains to be sensitive to carbapenems; also, the majority of them were sensitive to amikacin (93.2%) and cefoperazone/sulbactam (71.7%) [43]. Kot et al. revealed that 73.3 % of MDR *K. pneumoniae* strains were ESBL producers [43]. Although we did not investigate the specific mechanisms of antibiotic resistance in MDR *Klebsiella* spp. strains, the observed high prevalence (almost 50%) of combined resistance to penicillin, cephalosporins, aminoglycosides, and fluoroquinolones suggests a probable production of ESBLs. Our results suggest that, in Lithuania, carbapenems and aminoglycosides might need to be in the preferred first-line empirical antibiotic regimen for VAP when ESBL-producing MDR *Klebsiella* spp. is suspected. The emergence of carbapenem-resistant strains of this pathogen has significantly reduced the treatment options available, leaving polymixins, aminoglycosides, glycylcyclines, aztreonem, or combination therapies as possibly viable alternatives. Newer betalactamase inhibitors associated with carbapenems, such as imipenen-relebactam in the RESTORE IMI 1 and 2 studies [45,46] improved all-cause mortality in vHAP/VAP and patients with APACHE II scores above 15, adding value to the therapy of critically ill patients with Enterobacteriales and *P. aeruginosa.* Relebactam has limited class D activity, but our cohort in Lithuania does not require activity against metalobetalactamases.

Bringing fresh knowledge to the field of VAP, we additionally compared the clinical features of mono- and polybacterial cases of VAP due to MDR *Klebsiella* spp. on ICU admission and at VAP onset. In monomicrobial VAP, compared to the polymicrobial one, on ICU admission, diabetes mellitus, obesity, prior hospitalization, and prior antibacterial treatment, especially with cephalosporins, were more frequent, whereas at VAP onset, the presence of septic shock and a high cardiovascular SOFA score were more frequent (≥3). Polybacterial cases were associated with a higher SAPS II score and impaired state of consciousness at VAP onset, compared to the monomicrobial ones. Most VAP patients had had chronic diseases, and some of them underwent surgery (perioperative antibiotic prophylaxis) that might lead to immunosuppression and increased likelihood of infections. Therefore, they might have received antibiotics within 90 days before the onset of VAP, specifically for this reason. The higher prevalence of chronic underlying diseases in monobacterial VAP and/or the impaired consciousness in polybacterial VAP was also reported in the earlier studies by Ferrer, Cilloniz, and Natarajan [20,28,47]. The associations between a decreased level of consciousness and the polybacterial origin of VAP may be partially explained by the loss of the gag reflex, which may increase the risk of aspiration, with selected resistant organisms among patients exposed to long hospitalization and antibacterial therapies [47,48]. The resistance phenotype of pathogens may also have an impact on patient mortality [49,50]. In our study, the trend of increased mortality in polybacterial *Klebsiella* spp. VAP cases was observed in both the MDR and XDR groups.

### Study Novelties and Limitations

There are some limitations in our study. First, in our analysis, the respiratory cultures were not quantitative, potentially leading to an under- or over-estimation of some pathogens. Second, it was a retrospective, single-centre cohort, and although the data were collected from ICUs of various profiles in the largest Lithuanian university hospital, our results may not represent other hospitals in the country. However, since critically ill patients were transferred from regional hospitals to the university ICUs where the study was conducted, we expect our findings to reflect the country’s VAP profile due to MDR *Klebsiella* spp. fairly exactly, and thus might be generalizable to Lithuania. Due to the retrospective nature of this study, the sample size was limited to the eligible cases available in the database, limiting the statistical power, with risk of a type II statistical error (statistical power 0.8 with the effect size 0.303). Therefore, due to the relatively small sample size, important numerical differences (20%) in all-cause mortality of monobacterial versus polybacterial episodes were considered as clinically relevant, although they did not reach statistical significance. In addition, a sub-group analysis of MDR *Klebsiella* spp. strains with different resistance profiles and co-infection with different pathogens could not be conducted. During the study period, our hospital did not routinely investigate molecular mechanisms of antibiotic resistance and did not use rapid tests to detect such resistance mechanisms. In addition, our results cannot be generalized for geographical areas with a high prevalence of carbapenemases, and further studies are required, with important implications for stewardship or preventive strategies. The implications for antimicrobial stewardship or VAP prevention are beyond the scope of this article, and we refer the readers to recent updates [2,51,52,53,54,55,56,57,58,59] addressing these important issues.

## 5. Conclusions

In a cohort of 86 adults with *Klebsiella* spp. VAP in Lithuania, the epidemiological features and outcomes differ between monobacterial and polybacterial episodes. The index case was a polymicrobial episode (two-thirds) in an adult above 65 years, hospitalized longer than 13 days, presenting an all-cause high 60-day mortality. Polymicrobial episodes were often associated with neurological dysfunction but with fewer vasopressor requirements. Observed epidemiological differences suggest the need for further research, with mono/polybacterial cases being analyzed separately. These findings may have implications for therapy, particularly in areas with high rates of MDR, for guiding antimicrobial stewardship in ICUs with a high prevalence of *Klebsiella* spp. VAP.

## Figures and Tables

**Figure 1 antibiotics-12-01056-f001:**
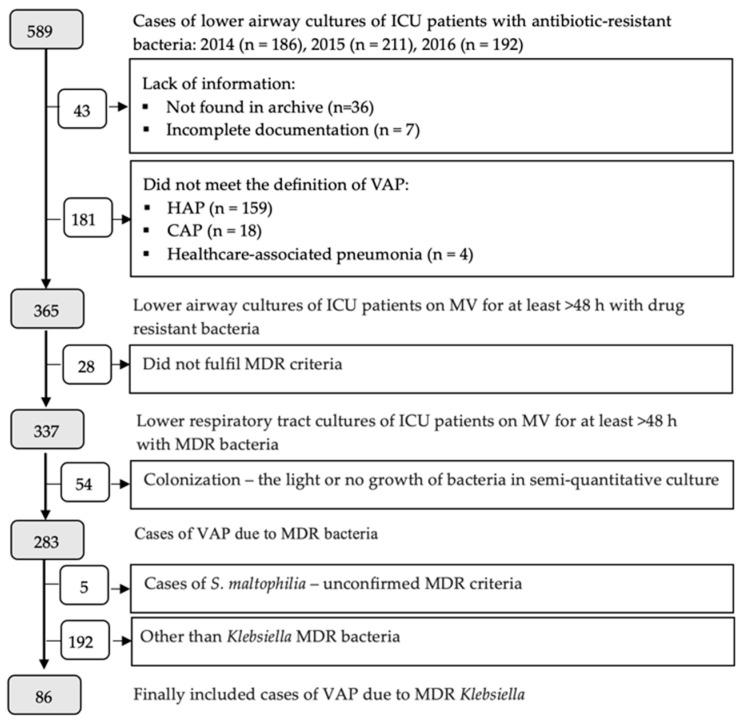
Case enrollment flow diagram. ICU—intensive care unit; HAP—hospital-associated pneumonia; HCAP—health-care-associated pneumonia; MV—mechanical ventilation; MDR—multidrug-resistant; VAP—ventilator-associated pneumonia.

**Figure 2 antibiotics-12-01056-f002:**
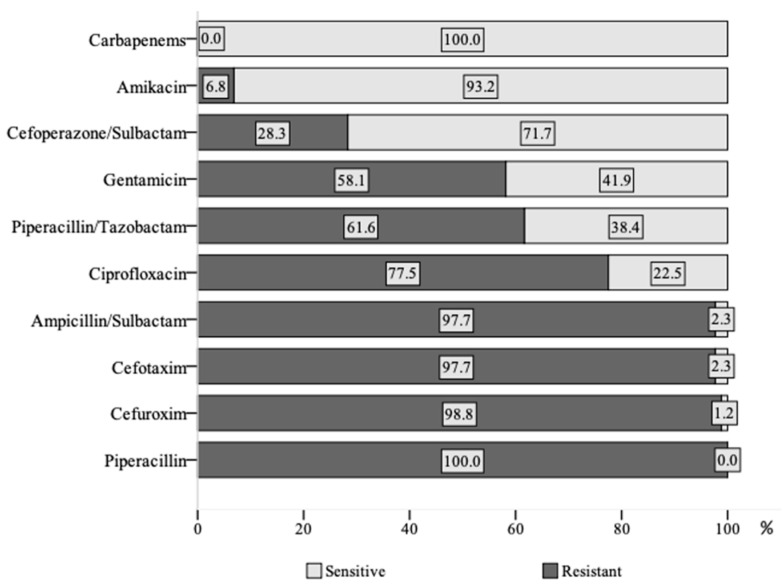
Antibacterial resistance of *Klebsiella* spp. strains.

**Figure 3 antibiotics-12-01056-f003:**
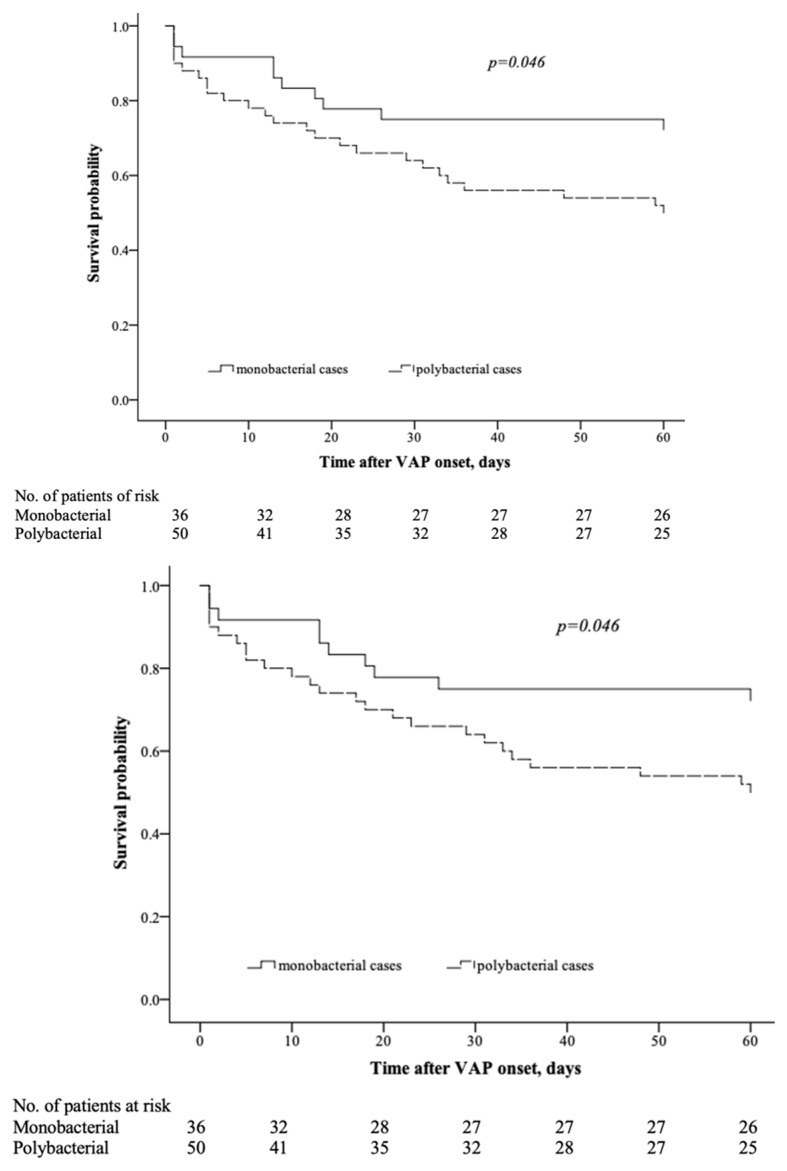
Kaplan–Meier survival curves for time to death in monobacterial vs. polybacterial VAP due to MDR *Klebsiella* spp. (censored at 60 days).

**Table 1 antibiotics-12-01056-t001:** Characteristics of mono- and polybacterial cases of VAP due to MDR *Klebsiella* spp.

Variable	VAP Origin		
	Monobacterial *n* = 36	Polybacterial *n* = 50	*p* Value
Age, years, median (IQR)	67.0 (55.0–75.0)	71.0 (57.5–76.5)	0.813
Sex, male, *n* (%)	32.0 (88.9)	44.0 (88.0)	0.899
Prior hospitalization within 90 days, *n* (%)	22 (61.1)	20 (40.0)	0.053
Hospitalisation to ICU from, *n* (%):			
▪ Community—ED ▪ Ward ▪ Other ICU	14 (38.9) 14 (38.9) 8 (22.2)	23 (46.0) 17 (34.0) 10 (20.0)	0.805
Admission, *n* (%):			
▪ Medical ▪ Surgical	11 (30.6) 25 (69.4)	27 (54.0) 23 (46.0)	0.148
Chronic illness, *n* (%):			
▪ Cardiovascular ▪ Respiratory ▪ Neurological ▪ Renal ▪ Diabetes ▪ Obesity *	31 (86.1) 28 (77.8) 4 (11.1) 5 (13.9) 18 (30.6) 11 (30.6)	38 (76.0) 35 (70.0) 3 (6.0) 11 (22.0) 5 (5.8) 4 (4.7)	0.245 0.421 0.392 0.340 0.01 0.007
The use of intravenous antibiotic within 90 days, *n* (%):	28 (77.8)	26 (52.0)	0.015
▪ Penicillins ▪ Cephalosporins ▪ Fluoroquinolones ▪ Aminoglycosides ▪ Carbapenems	12 (33.3) 24 (66.7) 4 (11.1) 4 (11.1) 3 (3.5)	12 (24.0) 8 (36.0) 4 (8.0) 3 (6.0) 2 (2.3)	0.341 0.005 0.715 0.446 0.645
Disease severity at VAP onset, median (IQR):			
▪ SAPS II score ▪ SOFA score	45.0 (35.25–51.0) 7.5 (6–9.75)	50.0 (40.5–60.75) 7.0 (5.0–8.25)	0.033 0.149
Organ failure at VAP onset, *n* (%):			
▪ SOFA respiratory ≥ 3 ▪ SOFA cardiovascular ≥ 3 ▪ SOFA neurological ≥ 3	28 (77.8) 16 (44.4) 8 (22.2)	36 (72.0) 5 (10.0) 26 (52.0)	0.545 <0.001 0.005

ED: emergency department; ICU: intensive care unit; IQR; interquartile range; IV: intravenous; SAPS II: Simplified Acute Physiology Score II; SOFA: Sequential Organ Failure Assessment; VAP: ventilator-associated pneumonia; * Obesity: body mass index over 30 kg/m^2^.

**Table 2 antibiotics-12-01056-t002:** All-cause 60-day and ICU mortality of mono- and polybacterial cases of VAP due to MDR *Klebsiella* spp.

Variable	60-Day Mortality		
	VAP Origin		*p* Value
	Monobacterial, *n*/Total (%) *	Polybacterial, *n*/Total (%) *	
All samples	10//36 (27.8)	25/50 (50.0)	0.039
Severity at VAP diagnosis			
▪ SOFA < 8 ▪ SOFA > 7	3/20 (15.0) 7/416 (43.8)	15/33 (45.5) 10/17 (58.8)	0.023 0.387
Appropriate treatment and severity at VAP diagnosis			
▪ SOFA < 8 ▪ SOFA > 7	2/16 (12.5) 6/15 (40.0)	10/26 (41.7) 6/13 (46.2)	0.049 0.743
Antibacterial resistance profile of *Klebsiella* spp. strains			
▪ MDR ▪ XDR	7/19 (36.8) 3/17 (17.6)	12/21 (57.1) 13/28 (46.4)	0.199 0.051

ICU: intensive care unit; MDR: multidrug-resistant; SOFA: Sequential Organ Failure Assessment; XDR: extensively drug resistant; VAP: ventilator-associated pneumonia. * *n*: number of deceased patients in the subgroup/total: total of the respective subgroup.

## Data Availability

The data presented in this study are available on request from the corresponding authors.

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
