# Peer review of "Clinical Features and Outcomes of VAP Due to Multidrug-Resistant Klebsiella spp.: A Retrospective Study Comparing Monobacterial and Polybacterial Episodes"

_antibiotics, 2023, doi:10.3390/antibiotics12061056_

Round 1

Reviewer 1 Report

The authors present an interesting article on a relevant topic in nosocomial infection. However, there are some aspects of the study that need to be improved and clarified. It is important for the authors to address all the issues raised so that the quality of the publication improves and can be considered for publication.

INTRODUCTION

Lines 52-66

This paragraph is an almost historical review of the Klebsiella genre that contributes little to the objective of the study. I kindly request the authors to reconsider this lengthy and of limited value paragraph.

Mayor issue

There is a gap in the hypothesis of the study because the presence of a mono or polymicrobial culture does not imply the origin of the infection. The origin of VAP/HAP can be due to micro aspiration, hematogenous, or adjacent foci, but this does not necessarily indicate whether it is mono or polymicrobial. Please, review it.

METHODS

Other focus description

There is no description of the previous or concomitant focus of infection. I suppose there is some previous infection because the patients have had previous antibiotic treatment. Please, clarify.

Mayor issue

There is no description of the sample size calculation or the B error calculated for the primary objective variable (mortality rate in log-rank test). Please, clarify.

Mayor issue

There is no description of why the authors choose the endpoint of 60 days and why there are no data on mortality at 28 days. Please, clarify.

RESULTS

The results are only presented in two tables and a graph. The tables do not follow the publication format; please adapt them to the format. The tables are very long and contain data that are not relevant to the study. If the study focusses on mortality from nosocomial infection, the severity of the patient on entering the intensive care unit may be of limited value if the infection occurred days before admission. Review the tables and simplify them to variables that are valuable for the study. Some of the results must be described in the text.

Mayor issue

The microorganism implicated in the polymicrobial infection is mandatory to describe accurately.

Mayor issue

The results are inconsistent for two reasons. First, in the K-M analysis, the mortality of patients with monobacterial infection remains constant from day 28 to day 60, suggesting that the focus of infection is evolving favourably despite the initial severity. However, patients with polymicrobial culture have lower infection severity and lower mortality. The second reason is that attributing any cause of mortality to infection in such a long post-infection period suggests that patients with polymicrobial infection die with the infection rather than from the infection.

Therefore, it is necessary to perform a multivariate analysis, a Cox regression, to adjust at least for SOFA and antibiotic treatment at 28 and 60 days.

DISCUSSION

Lines 221-228

The authors suggest that the polymicrobial infection may be more deleterious than the effect of each bacterium independently; this is in contrast to the less evident clinical signs. In addition, when multiple bacterial species are present at an infection site, they may compete for resources, such as nutrients and space, which can influence their growth and survival.

Lines 247-353

This paragraph is a review of the treatment of nosocomial pneumonia, but does not focus on discussing the topic of the article or its results. Modify it accordingly.

Lines 365-368

There is no explanation or hypothesis about the lower level of conscience and the source of infeccion. Please review it.

Author Response

We would like to thank the reviewer for his thorough review and the constructive comments. We have revised and amended the manuscript according to suggestions and we believe that it has been substantially improved.

Reviewer 2 Report

Somewhere at the beginning, define the acronym VAP.

The introduction seems extensive and challenging to follow; lines 52-58 could be shortened.

In Materials and Methods: Line 125, early antibacterial treatment equal empiric?

In results: Lines 139-141, “In association with MDR Klebsiella spp., one co-pathogen was found in 41 (80.0%), and two co-pathogens in 9 (20.0%) cases of polybacterial VAP. The most frequently isolated co-pathogens were P. aeruginosa and A. baumannii.

Question: Sensitivities for those co-pathogens?

Table 1: Mentions sepsis at VAP onset and septic shock at VAP onset.

Question: Any blood culture results?

In discussion: Lines 218-228, confusing since there is no data on co-pathogens.

In discussion: Lines 228-236, there is a missing reference (s).

In discussion: From Lines 247-354, the segment is too long and looks more like a review of therapeutic options for VAP.

Other comments:

No comments on X-ray findings.

No comments on molecular mechanisms of resistance.

No comments on rapid tests to detect mechanisms of resistance.

No comments on the prevention of VAP.

No comments on antimicrobial stewardship.

Please review and comment on the following:

Kadri SS. Clin Infect Dis 2018; 67:1803.

Kadri SS. Lancet Infect Dis 2021; 21:241.

Finally: The study title “Polymicrobial origin of VAP due to MDR Klebsiella is associated with higher late mortality” leads the reader to believe they will find a list of all other co-pathogens and their sensitivities, alas, only one line.

Author Response

We would like to thank the reviewer for his thorough review and the constructive comments. We have revised and amended the manuscript according to your suggestions and we believe that it has been substantially improved.

Round 2

Reviewer 1 Report

The authors have substantially improved the manuscript following the revision, but I still believe there are two unresolved aspects.

First. The patient inclusion diagram is very enlightening and provides valuable information, but it is necessary to calculate the study's power (Type II error) to precisely determine the margin of error attributable to the sample size limitation.

And Sencondly

It is essential that you duplicate Table 2 and Figure 2 (they can be included as a composite figure) with the information on 28-day mortality.

Author Response

-
